# Low Rates of Intrapulmonary Local Recurrence After Laser Metastasectomy: A Single-Center Retrospective Cohort Study of Colorectal Cancer Metastases [note 1]

**DOI:** 10.3390/cancers17040683

**Published:** 2025-02-18

**Authors:** Ahmad Shalabi, Sundus F. Shalabi, Thomas Graeter, Stefan Welter, Ahmed Ehab, Jonas Kuon

**Affiliations:** 1Thoracic and Vascular Surgery Department, SLK Lungenklinik Löwenstein, 74245 Löwenstein, Germany; 2Faculty of Medicine, Arab American University, Jenin P.O. Box 240, Palestine; sundus.shalabi@aaup.edu; 3Department of Thoracic Surgery, Lungenklinik Hemer, 58675 Hemer, Germany; stefan.welter@lkhemer.de; 4Pulmonary Medicine Department, SLK Lungenklinik Löwenstein, 74245 Löwenstein, Germany; 5Pulmonary Medicine Department, Mansoura University, Mansoura 35511, Egypt; 6Department of Thoracic Oncology, SLK Lungenklinik Löwenstein, 74245 Löwenstein, Germany

**Keywords:** laser-assisted pulmonary metastasectomy, colorectal cancer, local recurrence, spread through air spaces (STAS)

## Abstract

Pulmonary metastasectomy for colorectal metastases is a well-established intervention with curative intent that is performed in a lung parenchyma-sparing technique where possible. Among their numerous advantages, lung parenchyma-sparing resections entail an increased risk for recurrence at the resection site intrapulmonary. Laser-assisted pulmonary metastasectomy, which is the most lung parenchyma-sparing technique, causes the vaporization and coagulation of lung tissue at the resection site. We evaluated the local intrapulmonary recurrence rate after the laser-assisted pulmonary metastasectomy of 139 nodules for colorectal metastases in the pursuit of a lung parenchyma-sparing resection with a low risk of local recurrence at the resection site.

## 1. Introduction

It is estimated that up to 50% of patients with colorectal cancer (CRC) will develop distant metastases [1]. Depending on the stage of the primary tumor (I–III) at the time of resection, the cumulative rate of metachronous distant metastases is 6.4–48%, and the lung is the most common extra abdominal location of distant metastases [2,3]. Pulmonary metastasectomy (PM) is the second most frequent operative intervention in thoracic surgery, accounting for 15–50% of the workload in European thoracic surgery departments [4]. PM for colorectal metastases is a widely accepted and practiced intervention that prolongs survival in well-selected patients [5,6,7,8]. Despite recent advances in the field of chemotherapy and immunotherapy, PM remains an option with the potential to cure [5,9].

In light of the global consensus to resect pulmonary metastases in a lung parenchyma-sparing technique, wedge resection (using stapling devices) is currently considered the standard technique in PM for CRC [4,10,11]. However, in the case of multiple metastatic nodules or more centrally located nodules, wedge resection nor parenchyma-sparing resection would not be plausible [4,12]. Moreover, the occurrence of new metastatic nodules or lymph node metastases after PM, or recurrence at a previous resection site, ranges from 16 to 53% and could be as high as 76% in CRC [13,14,15]. Repetitive anatomical or wedge resections here would be costly in terms of lung function and quality of life. Neodymium:yttrium aluminum garnet (Nd:YAG) laser-assisted pulmonary metastasectomy (LPM) is a relatively recent innovation that was introduced to promote parenchyma-sparing PM and to enable the resection of a higher number of nodules or eventually repetitive resections [4,11,16]. According to Rolle et al., wedge resection using staplers results in 7 times more parenchyma sacrifices than LPM [17].

Local recurrence at the surgical site following limited parenchyma-sparing resections is a problem of significance that could influence prognosis [11,18,19,20,21,22]. Although it is well documented, its underlying mechanisms, especially in microscopically complete resection (R0 [23]), are not fully understood [12,19]. Local recurrence has been associated with tumor histo-biology, the size of the nodule, the amount of undetected micro metastases spread through air spaces (STAS) and, most importantly, the completeness of resection [18,19].

The use of a 1320 nm Nd:YAG laser causes the vaporization, carbonization and coagulation of lung tissue at the resection line, thereby theoretically increasing safety margins around resected nodules. Rolle et al. and Franzke et al. have been able to demonstrate this [24,25]. The objective of this retrospective single-center cohort study was to evaluate the rate of local intrapulmonary recurrence after LPM. We hypothesize from our personal experience, that the rate is low compared with other resection techniques. The study results will be compared with the current literature. The primary outcome was the rate of local intrapulmonary recurrence at the surgical margin. Secondary outcome was overall survival (OS) with a focus on local recurrence. Furthermore, we aim to identify risk factors for local intrapulmonary recurrence at the surgical margin.

## 2. Patients and Methods

### 2.1. Patients and Data Collection

Patients were identified and selected using the surgical coding and documentation. A list of patients with pulmonary metastases of colorectal origin with the intraoperative code ‘laser-assisted pulmonary metastasectomy’ was extracted from the hospital’s electronic documentation system. A retrospective analysis of these patients based on their medical and radiological records was performed. Patients with macroscopic residual disease (R2), repeated operations for local recurrence at the surgical margin, no evidence of CRC metastases in the resection specimen, missing follow-up or follow-up less than 24 months were excluded. In patients were another resection method was used for metastasectomy in addition to a laser, only the site of the laser resection was included and evaluated for local intrapulmonary recurrence. By applying these criteria; we identified 49 patients who underwent, consecutively between January 2010 and December 2018, ‘laser-assisted pulmonary metastasectomy’ at Löwenstein Lung Medical Center, (SLK Lungenklinik Löwenstein, Löwenstein, Germany). LPM is the local standard technique at Löwenstein Lung Medical Center.

In all of these 49 patients, the primary malignancy had been successfully controlled at the time of PM. Some patients had other non-pulmonary metastatic nodules that were already controlled or planned for curative treatment. The indication for PM or repeated PM (new nodules) in the case of disease recurrence was individually discussed for each patient. The follow-up was performed by the surgical team in the outpatient clinic or by the patient’s oncologist or pulmonologist. Follow-up was performed with computed tomography (CT) of the chest at 3-, 6- or 12-month intervals. All CT scans were reviewed by a board-certified radiologist or by the surgical team itself to assess for local intrapulmonary recurrence at the resection lines.

The following parameters were collected: age, gender, the stage of primary disease, disease-free interval (DFI), the use of chemotherapy, the number of nodules and their size and location, lymph node metastases, surgical approach (VATS vs. anterolateral thoracotomy), the completeness of resection (R0-2), complications, mortality, follow-up period and local recurrence at the surgical margin. The DFI was defined as the interval between the resection of colorectal cancer and the first detection of pulmonary metastases. The DFI was considered to be zero in cases where pulmonary metastases developed synchronously with the primary malignancy. Nodule size was defined as the size measured on the gross pathological examination. The completeness of the resection (R status), used for analysis, was the one given by the pathologist. When tumor cells reached the coagulation zone, Rx was used for analysis. Local recurrence at the surgical margin was defined as postoperative tumor development at the resection line (coagulation zone) detected on a chest CT scan by the radiologist or by the surgical team or confirmed pathologically after repeated resection.

### 2.2. Operative Procedure

Limax^®^ (KLS martin group, Tuttlingen, Germany), a diode-pumped Nd:YAG laser, was used in all PMs. LPMs over the period of nine years were performed by different surgeons. All of them were trained and strictly followed the hospital’s standardized operational procedure, taking a 5 mm safety margin of healthy lung tissue with each nodule. The operative approach (VATS or anterolateral thoracotomy) was set by the surgical team after thorough preoperative planning based on the contrast-enhanced chest CT scan, taking the patients’ functional status and fitness into consideration. In cases of bilateral disease, staged resections were performed within 6–8 weeks. All LPMs were performed under general anesthesia, with single-lung ventilation in a lateral decubitus position. Macroscopic complete resection was performed for each nodule.

### 2.3. Statistical Analysis

A Cox proportional hazard model was used to investigate the factors associated with an increased risk of local recurrence at the surgical site after laser-assisted pulmonary metastasectomy in univariable and multivariable analyses. All *p* values reported are of two-sided tests, and the significance level was set to less than 0.05. The statistical significance of each factor’s effect on local recurrence was assessed using the logrank test. All analyses and plots were generated using the ‘survival’ and ‘survminer’ packages in R^®^ version 4.0.3.

## 3. Results

### 3.1. Patients’ Characteristics

A total of 139 metastatic colorectal nodules were found and analyzed from the 49 patients included in this study. On average, each patient had 2.7 (1–8) nodules. In total, 14 patients had a single nodule only. Among the cohort, 33 patients (67.3%) were males with a mean age of 66.7 years, and 16 patients (32.7%) were females with a mean age of 61.8 years. The mean age of the whole cohort at the time of PM was 65.2 years (range/SD: 27–79 years ± 11.8). Baseline data per patient are depicted in Table 1. Four operations were performed thoracoscopically, while the rest of the operations were performed via anterolateral thoracotomy. The patients included were observed for a mean of 46.1 ± 19.7 months postoperatively.

### 3.2. Local Recurrence and Associated Factors

The 49 patients included in our study had a total of 139 nodules, as described in Table 2. After a minimum observation of 2 years postoperatively, local recurrence at the surgical margin was detected at seven resection sites (5%) in five patients (10.2%). Local recurrence occurred at these sites after 19.0 ± 7.4 months postoperatively. All five patients with local recurrence were reoperated, and local recurrences at the surgical margin were resected.

Local recurrence was influenced significantly by microscopic incomplete resection (hazard ratio (HR): 5.45, 95% CI: 1.057–28.12%, *p* = 0.023); this is clearly seen in Figure 1. Nodules that developed local intrapulmonary recurrence were significantly larger; the average size for nodules with local recurrence was 13.7 ± 3.77 mm (median = 12), whereas the average size of nodules without local recurrence was 9.5 ± 6.1 mm (median = 8) (*p*-value = 0.024, Welch’s two-sample *t*-test). This is plotted in Figure 2.

Other factors studied such as the number of nodules, laterality of the diseases, age of the patient, gender and DFI did not influence local recurrence at the surgical site.

### 3.3. Survival Analysis

Survival-related data could be collected from 46 out 49 patients; 11 (23.9%) patients died, and 35 (76.1%) were then censored. The 3- and 5-year OS of all patients was 83,7% and 53.1%. The median survival for patients who had local recurrence was inferior (65 months) compared to those who did not develop local recurrence (74 months). This difference, however, was not significant to predict survival (HR: 6.125, 95% CI: 0.60–61.87%, *p* = 0.125). Of the factors included in our analysis, only the size of the nodule affected survival. Patients with a maximum nodule size of less than 12 mm in diameter had better OS than those with larger lesions (HR: 0.3271, 95% CI: 0.1265–0.846, *p* = 0.018, Figure 3).

## 4. Discussion

Although the PulMiCC trial was highly anticipated from oncologists as well as surgeons to be the first randomized controlled trial on PM in CRC, it was unfortunately stopped due to low recruitment. Nevertheless, this trial was published in 2019. In total, 65 patients with CRC pulmonary metastases were randomized either to PM or to active monitoring [26]. The authors reported a five-year survival of 38% (23–62%) after PM and 29% (16–52%) in the well-matched controls. Although this trial remains underpowered to prove the benefit of PM and show its superiority, it serves as a verification of similar five-year survival rates previously reported in the PM literature, thereby indirectly showing PM superiority when proper patient selection is applied. Adopting a lung parenchyma-sparing PM technique is critically important considering the possible need for further future resections for bilateral disease or pulmonary recurrence. Here, LPM presents itself capable to resect a higher significant number of pulmonary metastases, reducing the need for a lobectomy without a negative influence on survival [16,27]. Moreover, a recent long-term follow-up showed total lung function recovery after LPM [28].

The inability to completely resect all metastatic nodules is widely considered a contraindication to PM [10]. Nearly all the PM literature demonstrated that the complete resection of metastases is associated with better outcomes [10,19,29,30,31]. Regardless of the different factors proven to increase the risk of local recurrence at the surgical margin, local recurrence is the result of incomplete resection on a microscopic level. Thus, local recurrence is an adverse event with a negative influence on survival [11,32,33]. Hence, this study aimed to assess the local recurrence rate at the surgical margin as the primary outcome and to identify potential risk factors that may influence local recurrence or survival as secondary outcomes.

Our study included 139 metastatic pulmonary nodules from 49 patients who underwent LPM with a curative intent. After a minimum postoperative follow-up of 24 months, seven nodules (5%), in five patients (10.2%) showed local recurrence at the surgical margin after 19.0 ± 7.4 months postoperatively. A safety margin of about 5 mm was maintained for each nodule to achieve the clinical completeness of the resection. The completeness of resection (R0) is considered the key factor for avoiding local recurrence [22,34]. After LPM, it is challenging to confirm complete resection histopathologically if the resection margin is not doubtlessly clear. This is a consequence of the vaporization along the resection line and the pronounced coagulation zone. If tumor cells reach the coagulation zone at the edge of the specimen, the pathologist labels it Rx ‘undetermined’, even though another 5–10 mm of vaporization and a coagulation zone remaining in the patient should be added. Therefore, the completeness of the resection after LPM should be clinically evaluated [22,25,35]. In this study, we had 46 nodules (33%) with Rx resection status. Five of these (10.8%) showed local recurrence at the resection line compared to only two (2.1%) local recurrences for R0 nodules (*p* < 0.023, logrank test) (Figure 1). This indicates that despite vaporization and coagulation zones on both sides of the resection line, some Rx resections were in fact incomplete (R1). But even when a complete resection was histopathologically confirmed (R0), local recurrence occurred. Although partially understood, this is a known phenomenon in the literature, especially in metastatic nodules of colorectal origin. Tumor spread through air spaces (STAS) is a unique means of tumor spread and growth in lung tissue, predominantly in metastases of colorectal origin [20]. Several studies have found STAS to be a risk factor for local intrapulmonary recurrence despite histopathologically complete resection (R0) [18,19,20].

Our analysis showed that local recurrence significantly correlates with nodule size. The average size for nodules with local recurrence was 13.7 mm, whereas the average size of nodules without local recurrence was 9.5 mm (*p* = 0.024). This is in accordance with the published literature on local recurrence after non-laser PM [20,21]. Metastasis size also affected OS. A significant cut-off point was found at 12 mm. Nodules larger than 12 mm lowered the patients’ probability of survival by 67.3% (*p* = 0.018, Figure 3) compared to patients with smaller nodules. It was demonstrated by Welter et al. that the presence of aggressive patterns of local metastasis growth increase with metastasis size, and the distance of STAS cells found in the surrounding lung tissue also increase with nodule size [18,20]. Therefore, increasing the safety margin for larger tumors is recommended. Nelson et al. affirmed that the surgical margin should be as large as the metastasis diameter to keep the recurrence rate low after wedge resection with staplers [21].

Three-year and five-year survival rates for all patients were 83.6% and 53.0%, respectively. The median survival for patients who had local recurrence was 65 months, whereas patients who did not develop local recurrence had a median survival of 74 months. This difference, however, was not significant to predict survival (*p* = 0.125) probably because patients who developed local recurrences were reoperated (the recurrence was resected).

Video-assisted (VATS) LPM was performed in four of our patients only. None of them had local recurrence at the surgical margin. The number was too small for further subgroup analysis. This approach for lesions located in the periphery of the lung might be a reliable alternative to the open approach. Some positive experiences have already been published [36,37].

### Limitations

This study had some limitations, so caution is required when interpreting the results. Nevertheless, these limitations should not impede the main finding that the local recurrence rate at the surgical margin after LPM with colorectal origin is low. This study was retrospective and included a relatively small number of patients and nodules. Surgical resections (LPM) in this study were performed by different surgeons, and the specimens were reviewed by different pathologists, thus raising the margin of human error. In total, 35 patients censored in a population of 49 patients is a high number that weakened the statistical analysis regarding survival. Risk factors for local intrapulmonary recurrence were incomplete because we only included R0/Rx and we were not able to define the safety distance of every removed metastasis. Adding a low local recurrence rate to this setting weakened and masked the effect of the examined factors possibly associated with increased risk for local recurrence at the surgical margin. The exact timeline of adjuvant therapy is not well known for each patient; thus, the possibility of underestimating undetected local recurrence under therapy cannot be excluded.

## 5. Conclusions

The local recurrence rate at the surgical margin after LPM is low when a pathologic complete resection can be achieved. Larger nodules require larger safety margins.

## Figures and Tables

**Figure 1 cancers-17-00683-f001:**
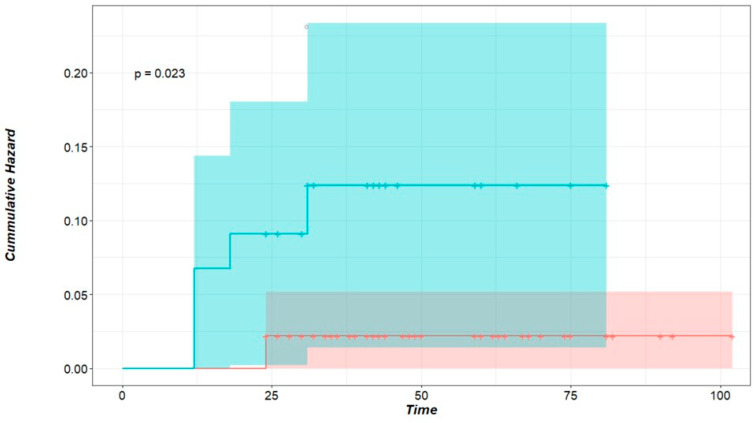
Cumulative hazard of local intrapulmonary recurrence for R status. Time in months, green curve: Rx nodules representing incomplete microscopic resection, red curve: R0 nodules representing nodules with clear-cut margins. *p* < 0.023 (logrank test).

**Figure 2 cancers-17-00683-f002:**
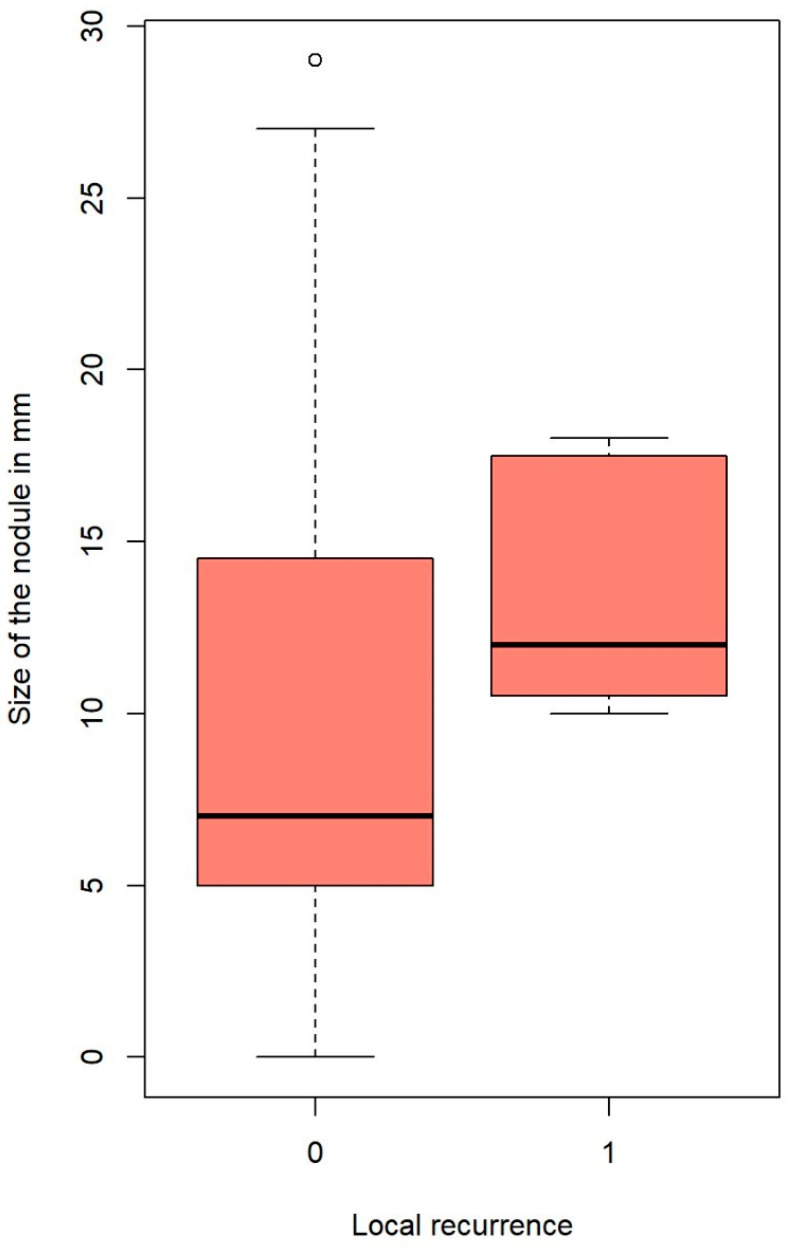
Nodule’s size and local recurrence. The outlines of the boxes represent the first and third quartiles. The vertical line inside the boxes represents the median, and the whiskers go from each quartile to the minimum and maximum values.

**Figure 3 cancers-17-00683-f003:**
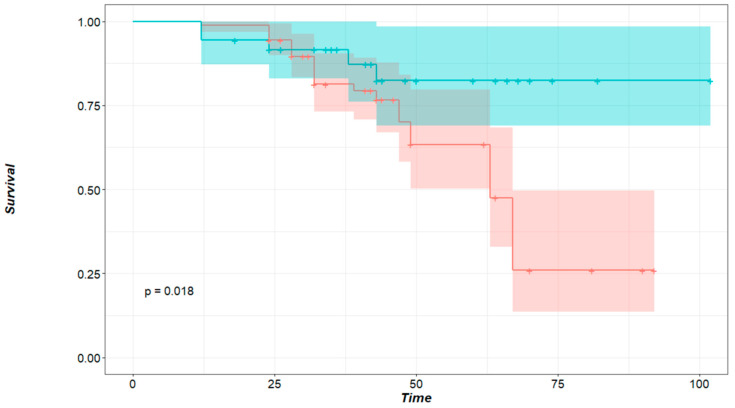
Survival probability in regard to the size of the nodule. Time in months, green curve: patients with nodules smaller than 12 mm, red curve: patients with nodules larger or equal to 12 mm in size. *p* = 0.018 (logrank test).

**Table 1 cancers-17-00683-t001:** Patients’ characteristics.

		Patients/Percent	Local Recurrence Patients (Percent)
Gender	Male	33 (67.3%)	4 (12.1%)
	Female	16 (32.6%)	1 (6.2%)
Age group	≤35 Years	2 (4.8%)	0
	35–55 Years	5 (10.2%)	0
	≥55 Years	42 (85.7%)	5 (11.3%)
Disease spread	Unilateral	25 (51.0%)	2 (8.0%)
	Bilateral	24 (49.0%)	3 (12.0%)
DFI in months (n = 45)	<36	35 (77.8%)	3 (8.5%)
	≥36	10 (22.2%)	2 (20%)
Extrapulmonary invasion	Yes	0	0
	No	49	5 (10.2%)
Single nodule	Yes	14	2 (14.2%)
	No	35	3 (8.5%)
Surgical access	VATS	4	0
	Thoracotomy	45	5 (11.1%)
Lymph node metastasis	Yes	1	0
	No	48	5 (10.4%)
Age	mean	65.2	
	SD	11.8	
	SEM	1.68	
DFI in months (NA = 4)	Mean	15.8	
	SD	18.5	
	SEM	2.6	
Number of nodules per patient	Mean	2.7	
	SD	2.5	
	SEM	0.37	
Post-observation time in months	mean	46.0	
	SD	19.7	
	SEM	2.46	

Legend 1: baseline data per patient. DFI: disease free interval; SD: standard deviation; SEM: standard error of mean.

**Table 2 cancers-17-00683-t002:** Nodule characteristics and recurrence.

		Number (Percent)	Local Recurrence/Nodule (Percent)
Gender	Male	92 (66.2%)	6 (6.5%)
	Female	47 (33.8%)	1 (2.1%)
R status	R0	93 (66.9%)	2 (2.1%)
	R_x_	46 (33.1%)	5 (10.9%)
Size	<12 mm	94 (67.6%)	3 (3.2%)
	≥12 mm	45 (32.4%)	4 (8.9%)
Number of nodules	Single	14 (10.1%)	2 (14.3%)
	Multiple	125 (89.9%)	5 (4.0%)
Laterality	Ipsilateral	38 (27.3%)	2 (5.3%)
	Bilateral	101(72.7%)	5 (5.0%)
Node location	LL	68 (48.9%)	2 (3.0%)
	ML	16 (11.5%)	1 (6.2%)
	UL	55 (39.6%)	4 (7.3%)
Total		139	7 (5.0%)

Legend 2: R_x_, incomplete microscopic resection; R0, at least one layer of healthy tissue on the surface of the resected metastasis; UL, upper lobe; ML, middle lobe; LL, lower lobe.

## Data Availability

The dataset used and analyzed during the study is available from the corresponding author on request.

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
