# Peer review of "Low Rates of Intrapulmonary Local Recurrence After Laser Metastasectomy: A Single-Center Retrospective Cohort Study of Colorectal Cancer Metastases [Author-notes fn1-cancers-17-00683]"

_cancers, 2025, doi:10.3390/cancers17040683_

Round 1

Reviewer 1 Report

Comments and Suggestions for Authors

Congratulations to you on your brilliant clinical study of reducing local recurrence over the resection line after pulmonary metastasectomy using Nd-YAG laser. The entire manuscript was well written with good language and layout. The results seem encouraging for the future clinical practice in terms of treating pulmonary metastases. However, there are several defects including study design causing the consideration of acceptance for publication doubtful. My comments are as follow.

(1) How did you differentiate local recurrence from scar formation over the resection line, if you did not perform repeat resection ? Did you perform every hypertrophic scar found on CT scan ?

(2) Continuation of question(1) ,  Would there be any possibility that local recurrence over the resection line not obviously shown on the CT scan because of under control by adjuvant chemotherapy ? That will be an underestimation.

(3) There was no control group in this study .

(4) Half of the references are over ten years old and I suggest replacement.

Author Response

For research article

Low rates of intrapulmonary local recurrence after laser metastasectomy. A single center retrospective cohort study of colorectal cancer metastases.

Response to Reviewer 1

Summary

Thank you very much for taking the time to review this manuscript and for your motivating words. Please find the detailed responses below and the corresponding corrections highlighted in the re-submitted files.

Comment 1: [How did you differentiate local recurrence from scar formation over the resection line, if you did not perform repeat resection? Did you perform every hypertrophic scar found on CT scan?]

Response 1: Thank you for your question pointing this out for discussion. We were able you differentiate local recurrence from scar formation through sequential follow-up CTs. The initial postop inflammatory changes would form a scar within months. Afterwards, this scar should not grow or widen, otherwise the suspicion of recurrence is raised. The following CT section for the same patient illustrates this process over time. All patients that we reoperated with this clinical / radiological suspicion confirmed local recurrence at the resection margin pathologically.

This method of differentiation is also described by (Prisciandaro et al., 2022), however, CT-guided biopsy is occasionally doable and reasonable.  

Ref.

Prisciandaro, E. et al. (2022) ‘Impact of the extent of lung resection on postoperative outcomes of pulmonary metastasectomy for colorectal cancer metastases : an exploratory systematic review’, 14(7), pp. 2677–2688. doi: 10.21037/jtd-22-239.

Comment 2: [Continuation of question(1) ,  Would there be any possibility that local recurrence over the resection line not obviously shown on the CT scan because of under control by adjuvant chemotherapy ? That will be an underestimation..]

Response 2: Thank you again for raising this point by your question. Theoretically, this is of course possible. But, this can be neither proven or denied or even tested. As the exact timeline of adjuvant therapy for each patient is not known, this will be added to the limitation of the study (marked in red). 

Comment 3: [There was no control group in this study.]

Response 3: We agree that a control group would be informative, but this beyond the aim of this work, which is a descriptive and not a comparative study. Moreover, and besides the sample size issue that you thankfully well described, an appropriate and matched control group from our institution is hard to generate as we are a reference center for

 laser-assisted metastasectomy where is our local standard approach for metastasectomy.

Comment 4: [Half of the references are over ten years old and I suggest replacement.]

Response 4: Thank you for pointing this out, recent references has been added and older ones updated. Nevertheless, some cardinal references that report local recurrence at the resection margin are irreplaceable.

Reviewer 2 Report

Comments and Suggestions for Authors

This single institution, retrospective cohort study describes the efficacy of laser-assisted resection of lung metastasis from colorectal cancer in terms of rate of local recurrence. 

To the best of my knowledge, the study, altough retrospective, is one of the largest on this relevant topic.

The paper is very well written!

Results are clearly presented and discussed at the light of the available evidences. Moreover, its limitations are correctly aknowledged.

Nevertheless, in my opinion the paper needs minor changes in order to meet the requirements for publication.

In particular:

1.Although in “Introduction” has been stated that laser-assisted resection is indicated "in case of multiple metastatic nodules or more centrally located nodules", when "wedge would not be plausible", I think that elegibility criteria for LPM adopted by the treating Instidtution should be clearly described in "Material and methods". This will be usefull in excluding the chance of selection bias, which are notoriously a problem in retrospective study

2. Were the selected patients consecutively treated?

3. The sample size is quite limited and realistically does not allow any formal comparison between different treatment. However, I think that the inclusion of an appropriate control group from the same Institution should be considered. The recurrence rate in this group will be much more informative than that obtained from the available literature and reported in “Introduction” (“as high as 76%”)

Author Response

For research article

Low rates of intrapulmonary local recurrence after laser metastasectomy. A single center retrospective cohort study of colorectal cancer metastases.

Response to Reviewer 2

Thank you very much for taking the time to review this manuscript and for your motivating words. Please find the detailed responses below and the corresponding corrections highlighted in the re-submitted files.

Comment 1: [Although in “Introduction” has been stated that laser-assisted resection is indicated "in case of multiple metastatic nodules or more centrally located nodules", when "wedge would not be plausible", I think that eligibility criteria for LPM adopted by the treating Instidtution should be clearly described in "Material and methods". This will be useful in excluding the chance of selection bias, which are notoriously a problem in retrospective study]

Comment 2: [Were the selected patients consecutively treated?]

Response: Thank you for pointing this out, this is a valid point. This is to answer question one & two.

This is one of the important indications/ applications of laser-assisted metastasectomy but not the only. This cohort was generated retrospectively by recruiting patients operated (laser-assisted metastasectomy by colorectal cancer) consecutively from 2010 till 2018 without skipping. In our center, metastasectomy is performed laser-assisted unless it is very central that a lobectomy is not avoidable. This detail is now better described (marked in red) in the revised manuscript.

Comment 3: [The sample size is quite limited and realistically does not allow any formal comparison between different treatment. However, I think that the inclusion of an appropriate control group from the same Institution should be considered. The recurrence rate in this group will be much more informative than that obtained from the available literature and reported in “Introduction” (“as high as 76%”)]

Response 3: We agree that a control group would be informative, but this beyond the aim of this work, which is a descriptive and not a comparative study. Moreover, and besides the sample size issue that you thankfully well described, an appropriate and matched control group from our institution is hard to generate as we are a reference center for

 laser-assisted metastasectomy where is our local standard approach for metastasectomy.

 Recurrence rates up to 76% mentioned in the introduction section are referring to disease   recurrence ipsilateral or bilateral or even in the lymph nodes. Local recurrence rate at the surgical margin after non-laser PM is relatively high, with published reports

ranging from 9 to 30%[1], [2][3].

Ref:

[1]        S. Shiono et al., “Histopathologic prognostic factors in resected colorectal lung metastases,” Ann. Thorac. Surg., vol. 79, no. 1, pp. 278–282, 2005, doi: 10.1016/j.athoracsur.2004.06.096.

[2]        J. H. Chung et al., “Impact of resection margin length and tumor depth on the local recurrence after thoracoscopic pulmonary wedge resection of a single colorectal metastasis,” J. Thorac. Dis., vol. 11, no. 5, p. 1879, 2019.

[3]        K. Franzke et al., “Pulmonary metastasectomy – A retrospective comparison of surgical outcomes after laser-assisted and conventional resection,” Eur. J. Surg. Oncol., vol. 43, no. 7, pp. 1357–1364, 2017, doi: 10.1016/j.ejso.2016.09.001.

Reviewer 3 Report

Comments and Suggestions for Authors

Dear Authors

Manuscript explains well and concludes that - the rate of local recurrence at the resection site after LPM for colorectal metastases is low. Complete resection is a positive predictor of survival without local recurrence. Microscopic complete resection with the addition of vaporization and coagulation at the resection margin seems to be sufficient to prevent local recurrence.

The following steps should provide more clear information for readers to enjoy it

1) Please add up-to-date references in the introduction section.

2) Keep separate sections in methods – Statistics analysis.

3) Table 1 - Patients characteristics - Age group - Patients / percent - 51 patients – Please correct it.

Author Response

For research article

Low rates of intrapulmonary local recurrence after laser metastasectomy. A single center retrospective cohort study of colorectal cancer metastases.

Response to Reviewer 3

Thank you very much for taking the time to review this manuscript and for your motivating words. Please find the detailed responses below and the corresponding corrections highlighted in the re-submitted files.

Comment 1: : [ Please add up-to-date references in the introduction section. ]

Response 1: Thank you for pointing this out, recent references has been added and older ones updated.

Comment 2: [Keep separate sections in methods – Statistics analysis.]

Response 2: We did, changes are marked with red in the revised manuscript.

Comment 3: [Table 1 - Patients characteristics - Age group - Patients / percent - 51 patients – Please correct it]

Response 1: Thank you for pointing this out, this was an unnoticed tipping error. Changes are marked with red in the revised manuscript.
